Identification of limiting climatic and geographical variables for the distribution of the tortoise Chelonoidis chilensis (Testudinidae): a baseline for conservation actions

Ruete Alejandro 1 aleruete@gmail.com
Leynaud Gerardo C. 2
1 Ecology Department, Swedish University of Agricultural Sciences , Uppsala , Sweden
2 Centro de Zoología Aplicada, Facultad de Ciencias Exactas Físicas y Naturales, Instituto de Diversidad y Ecología Animal, CONICET—Universidad Nacional de Córdoba , Córdoba , Argentina
Esler Karen
Electronic publication date: 2015 Oct 1
Publication date: 2015
Volume: 3
Electronic Location ID: e1298
Received 2015 Jul 14; Accepted 2015 Sep 16
Copyright: © 2015 Ruete and Leynaud
Copyright year: 2015
Copyright holder: Ruete and Leynaud
License: This is an open access article distributed under the terms of the Creative Commons Attribution License, which permits unrestricted use, distribution, reproduction and adaptation in any medium and for any purpose provided that it is properly attributed. For attribution, the original author(s), title, publication source (PeerJ) and either DOI or URL of the article must be cited.
License URL: https://creativecommons.org/licenses/by/4.0/

Keywords: Argentina, AUC, Bayesian inference, Bolivia, Chaco tortoise, Chelonoidis chilensis, Paraguay, Presence-only data, Protected areas, Species distribution model

Funding: The authors received no funding for this work.

==============================
Background. Just as for most other tortoise species, the once common Chaco tortoise, Chelonoidis chilensis (Testudinidae), is under constant threat across it distribution in Argentina, Bolivia and Paraguay. Despite initial qualitative description of the species distribution and further individual reports of new locations for the species, there is no description of the species distribution in probabilistic terms. With this work we aim to produce an updated predictive distribution map for C. chilensis to serve as a baseline management tool for directed strategic conservation planning.

Methods. We fitted a spatially expanded logistic regression model within the Bayesian framework that accounts for uncertainty on presence-only and generated pseudo-absence data into the parameter estimates. We contrast the results with reported data for the national networks of protected areas to assess the inclusion of the species in area-based conservation strategies.

Results. We obtained maps with predictions of the occurrence of the species and reported the model’s uncertainty spatially. The model suggests that potential suitable habitats for the species are continuous across Argentina, West Paraguay and South Bolivia, considering the variables, the scale and the resolution used. The main limiting variables were temperature-related variables, and precipitation in the reproductive period.

Discussion. Given the alarming low density and coverage of protected areas over the distribution area of C. chilensis, the map produced provides a baseline to identify areas where directed strategic conservation management actions would be more efficient for this and other associated species.

Introduction

Globally, turtles and tortoises are the most threatened group of vertebrates, with over half of all their species threatened with extinction (Van Dijk, Stuart & Rhodin, 2000; Turtle Conservation Fund, 2002). Exploitation and unregulated trade are the primary causes for sharp declines in many turtle species, with habitat loss and degradation also being major factors in widespread declines (Gibbon et al., 2000; Van Dijk, Stuart & Rhodin, 2000; Turtle Conservation Fund, 2002). Therefore, without directed strategic conservation planning, a significant portion of turtle diversity could be lost over the next century (Buhlmann et al., 2009).

The common Chaco tortoise, Chelonoidis chilensis (Testudinidae, Gray 1870), is mainly found in Argentina, but also in Bolivia and Paraguay (Cei, 1993; Cabrera, 1998). Although we still lack demographic evidence of population decline, C. chilensis is severely threatened by habitat degradation, poaching and illegal trade (Chebez, 2009). Thus, persistent threats and continuous habitat transformation lead to its categorization as Vulnerable by the IUCN (Tortoise & Freshwater Turtle Specialist Group, 2010) and its inclusion in the Appendix II of CITES.

An updated and refined distribution map of the species is required to serve as baseline for future conservation measures. Specifically, it is important to evaluate the contribution of the protected area network to the conservation of C. chilensis. However, despite initial qualitative and coarse descriptions of the species distribution (e.g., Waller, 1986; Buskirk, 1993; Richard, 1999) and further individual reports of new locations for the species (e.g., Cei, 1993; Gonzales, Muñoz & Cortéz, 2006; Fritz et al., 2012), there is no description of the distribution of the species in probabilistic terms. Species distribution models (SDMs) are a way of linking species occurrence data to environmental variables explaining and limiting the species distribution (Cassini, 2011). SDMs can further provide a spatial depiction of probabilities of occurrence rather than deterministic presence–absence maps. Because of the lack of systematic surveys at a national scale, the only data available for this species are presence-only records from museums and other literature. Given the great uncertainty inherent to this kind of data, state of the art Bayesian modelling techniques are required to account for this uncertainty and present it as part of the resulting species distribution map (Lobo, Jiménez-Valverde & Hortal, 2010).

With this work we aim to produce an updated predictive distribution map for Chelonoidis chilensis, based on geographical and bioclimatic explanatory variables and accounting for data uncertainty, as a baseline management tool for the conservation of the species. We aim to (i) gather as many records as possible for the species in Argentina, Paraguay and Bolivia; (ii) develop a probabilistic species distribution model using presence-only data that accounts for data uncertainty and spatial autocorrelation of the explanatory variables; and (iii) determine the inclusion of the species on protected areas by comparing model predictions to independent presence–absence datasets for the protected areas.

Methods

Data collection

In the current study the species is defined following Fritz et al. (2012), who concluded that Chelonoidis chilensis (Gray, 1870), C. donosobarrosi (Freiberg, 1973) and C. petersi (Freiberg, 1973) are the same species (i.e., C. chilensis). C. chilensis is a burrow-nesting species, found on sandy soils in scrublands or dry forests in the ecoregions of Monte and Chaco (Fig. 1; Cei, 1993; Cabrera, 1998), up to 1200 m.a.s.l. (Cerro Nevado, Mendoza; Richard, 1988). We collected confirmed observations of the Chaco tortoise dated 1950–2012 from the EMYSystem World Turtle Database (http://emys.geo.orst.edu/), and from scientific literature (Waller, 1986; Buskirk, 1993; Ergueta & Morales, 1996; Cabrera, 1998; Ernst, 1998; Richard, 1999; Gonzales, Muñoz & Cortéz, 2006; Fritz et al., 2012). We merged all reported observations in a GIS vector layer using QuantumGIS 1.8 (Quantum GIS Development Team, 2012). In case of overlap within 5 km we kept only the latest observation to avoid duplicated reports, and oversampling in densely populated areas. We also excluded three observations located close to Buenos Aires city because based on previous descriptions of the species distribution they most likely belong to translocated individuals. For a complete list of the 244 observations and corresponding sources, see Table S1. We arbitrarily defined the study area (Fig. 1) larger than the observed species distribution to include surrounding areas where the species is known to be absent. We excluded Chile from the study area because the Andean Mountain Range is a physical barrier the species cannot pass (i.e., highest observation at 1200 m.a.s.l.; Richard, 1988).

Figure 1 Maps of observations and a priori probability distribution of Chelonoidis chilensis.

Map of austral South America, showing (A) sites of confirmed observations of Chelonoidis chilensis (blue dots) and ecoregions where the species has been observed (coloured polygons); (B) a priori probability distribution of observations (colour scale) estimated from observation densities.

We obtained geographic and bioclimatic data from raster layers with 5 km resolution from two world databases (Hijmans et al., 2005; Hengl, 2009). The complete list of variables included in the study is presented in Table S2. We did not included land-use variables in the analysis because the data collected covers a wide temporal range (1950–2012), and the landscape has changed dramatically over this time period.

Modelling the species distribution

We implemented a Bayesian spatially expanded logistic (BSEL) model (Casetti, 1997; Congdon, 2003) to obtain the probability of occurrence at non-visited locations. Non-visited locations were randomly located with the same density as the observed locations (∼0.0004/km2). Given the nature of presence-only data, predicted probabilities combine the probability of the species being at the location, the probability of an observer being at the same location, and the probability of the observer finding the species (Lobo, Jiménez-Valverde & Hortal, 2010). The Bayesian approach allows us accounting for all three uncertainty sources on each observation, and displaying the model’s uncertainties spatially. We assume that occurrences at every non-visited location i are distributed according to a Bernoulli distribution Obsi∼Bernoullipi*, where pi* is an a priori probability distribution generated from confirmed observations (Fig. 1B). We generated the a priori probability distribution as a quadratic density kernel raster layer using the R package “splancs” (Rowlingson et al., 2013). By generating a prior distribution from the observations, we assume that the entire study region has been sampled for the species with the same intensity, which is a fair assumption given the map resolution and the time span of the study.

We then modelled observations Obsi according to a logistic model, Obsi ∼ Bernoulli (pi). The spatially expanded model (Casetti, 1997; Congdon, 2003) assumes that the effect of an explanatory variable on the response variable pi can vary among the observed locations. This assumption is particularly convenient when fitting species distribution models along large ranges, where the species can be locally adapted to e.g., temperature ranges (Turchin & Hanski, 1997; Nilsson-Örtman et al., 2013). The model was fitted using JAGS 3.0 (Plummer, 2013) through R (R Development Core Team, 2014). For further details on the modelling approach, see Appendix S1.

The final model presented (Table 1) is the result of a forward stepwise selection procedure based on the deviance information criterion (DIC), an information-theoretic criterion similar to Akaike’s information criterion (a.k.a. AIC), that is appropriate for Bayesian hierarchical modelling (Spiegelhalter et al., 2002). For further details on the selection procedure and all tested variables, see Appendix S1 and Table S2.

Table 1 Explanatory variables included in the final model.

Deviance Information Criterion (DIC) calculated stepwise from the null model (Table S2). Estimates of the effect parameters (δ) are extracted from the final model.

	DIC	δ¯ a	δ 95% CI	
Mean annual temperature	930.4	0.56	−4.68	5.14	
Max. temperature of warmest month	869.4	1.61	−2.05	5.86	
Temperature annual range	853.7	0.03	−2.25	2.12	
Precipitation of warmest quarter	824.5	−1.57	−2.35	−0.80	
Notes.

a Mode of the effect parameter.

Once the final model was obtained, we generated maps for the occurrence probability. We predicted occurrence probabilities for regularly distributed locations with the same resolution as the raster images for environmental variables (i.e., 5 km). We generated raster layers for the mode and for the length of the 95% credible interval (95% CI). The length of the 95% CI is a measure of precision ranging from 0 (precise) to 1 (imprecise).

Model evaluation

We calculated the AUC index with the SDMtools package for R (VanDerWal et al., 2012), contrasting predictions (Fig. 2A) against the a priori probability distribution (Fig. 1B). Then, we contrasted model predictions with two independent datasets of observations of Chaco tortoises in Argentinean and Bolivian protected areas (a similar dataset for Paraguay was not available). The first data set is mainly based on park rangers reports, and includes 144 Argentinean protected areas in the study area (Sistema de Información de Biodiversidad, SIB; Administración de Parques Nacionales, 2012). The second data set was put together in the framework of a doctoral thesis (Embert, 2007), and includes museum and field systematic collections for 38 Bolivian protected areas in the study area. The species was reported as present in 14 Argentinean and 3 Bolivian protected areas (Table S3).

Figure 2 Predictions of the species distribution model.

Maps showing (A) mode and (B) length of the 95% Credible Interval (CI) of probabilities of occurrence generated with the Bayesian Spatially Expanded Logistic model (BSEL). Blue lines show ecoregions delimitation for comparison with Fig. 1A.

Results

The final species distribution model obtained for Chelonoidis chilensis was mainly driven by temperature-related variables, but also included water availability in the reproductive period (i.e., precipitation in warmest quarter; Table 1). From this model, we generated maps displaying the probability of occurrence (Fig. 2A) and the model’s uncertainty. The model suggests that potential suitable sites of the species are continuous across Argentina, West Paraguay and South Bolivia, considering the variables, the scale and the resolution used. The model’s predictions generally overlap with published distribution maps for the species (Waller, 1986; Ernst, 1998; Richard, 1999; Administración de Parques Nacionales, 2012; Fritz et al., 2012) and with the ecoregions where the species has been described from (Fig. 1A). The model predictions’ accuracy is relatively high (AUC = 0.92). The uncertainty of the model was generally low (i.e., 95% CI length < 0.5, Fig. 2B) and is lower in areas where the occurrence probability is close to either 0 or 1 (Fig. S1). However, uncertainty is highest in areas with low density of observations (e.g., Bolivia).

The model has very low omission error (i.e., false negatives). In Argentina, the model predicts low occurrence probabilities (p < 0.5) for only one out of 14 protected area where the species has been reported (p = 0.45; Table S3.1). However, although the model predicted p < 0.3 for all protected areas in Bolivia (Fig. S2, Table S3.2), there were confirmed observations in three protected areas. Conversely, the commission error is high. That is, there is large discrepancy between positive predictions and the validation dataset, indicating potential false positives. Out of a total of 25 Argentinean protected areas where the model predicted p ≥ 0.5 the species was not reported in 12 of them.

Discussion

The model predicts a large and continuous area where Chelonoidis chilensis may be found, taking into account the selected variables, the scale of the study area and the resolution used. In general terms, temperature-related variables constrain the latitudinal and altitudinal range of the species, as it is expected for ectothermic species like amphibians and reptiles (Araújo, Thuiller & Pearson, 2006). Even more, precipitation-related variables constrain its range longitudinally. Altitude is certainly correlated to annual mean temperature, and this may be the reason why altitude did not improve the model fit when added to the full model. In the same way, leaf area index is expected to be highly correlated to precipitation in summer (i.e., the reproductive season), a variable that explains the adaptation of the species to dry environments (Waller, 1986; Buskirk, 1993; Ernst, 1998). However, correlation among variables per se was not a deterrent to test for variables together in a model because the aim of a predictive model is to capture and explain as much variability in the response variable as possible (Reichert & Omlin, 1997).

Model uncertainty and usage of predictions

An honest display of model uncertainties is crucial to evaluate and validate model predictions. We observed that higher uncertainty is expected on transition areas between high and low estimated probabilities or on poorly sampled areas (Figs. 2B and S1). In general, probabilities obtained for each pixel on the map have uncertainties associated to the observation events (Lobo, Jiménez-Valverde & Hortal, 2010), as well as to the model that generated those probabilities (Congdon, 2003; Clark & Gelfand, 2006). Model uncertainty maps complement the information contained on point estimate predictions, and should be displayed as yet another SDM result. Species distribution maps generated with low quality data (e.g., presence-only data) could be dangerously misleading if not accompanied with the corresponding uncertainty map. Too high or too widely spread uncertainty would also question the accuracy of the model, suggesting that more observations or alternative explanatory variables should be considered in the study. Also, uncertainty maps can be a valuable tool for designing field work efficiently. The researcher can then decide to focus future sampling effort either on areas with high uncertainty to validate the model or on areas with high probabilities of occurrence and low error to sample more efficiently.

For many practical applications, it is necessary to transform continuous maps to binary presence–absence maps assuming a (more or less) objective detection threshold (Liu et al., 2005; Jiménez-Valverde & Lobo, 2007). It is the researcher’s task to decide on which side of the detection threshold he/she wants the most of the model’s uncertainty. Liu et al. (2005) and Jiménez-Valverde & Lobo (2007) previously discussed that a threshold t = 0.5 is not always the best option, although it is often used. We observed that our model predictions have the highest uncertainty (widest 95% CI range) on regions where predicted occurrence probabilities are close to 0.5 (Fig. S1). Choosing t = 0.4 would leave higher uncertainties on values interpreted as presences. The opposite is also true for t = 0.6.

There are a few considerations to take into account when comparing the model predictions with the presence of the species on protected areas. First, it is important to consider the bias present on the distribution of protected areas. For example, commission error (i.e., false positive rate) is probably underestimated on the east of the species distribution (Espinal and Pampas ecoregions, <1% protected) if compared to the cover on the core distribution area (Monte and Chaco ecoregions, 3.7% protected) (Chebez, 2009) because of a heavily unbalanced distribution of protected areas (Fig. S2). Second, there are no data on the probability of detection or in the completeness of observation reports within the protected areas. Chelonidis chilensis is declared a “species of interest” by the National Parks Administration Office, requiring from its entire field staff to report observations of the species to the National Biodiversity Information System (Administración de Parques Nacionales, 2012). However, this is not necessarily the case for provincial, municipal or private protected areas. Therefore, the discrepancy between high predicted probabilities of occurrence with low uncertainty on protected areas where the species was not reported could be simply due to lack of reports or local extinction instead of model commission error.

Implications for conservation in protected areas

An important aspect for the conservation of C. chilensis that emerges from this study is the low density and total area of protected areas within the distribution of the species. Only 3.7% of Monte and Chaco ecoregions are under some form of protection (“Secretaria de Ambiente y Desarrollo Sustentable”). This value is far from the Aichi Biodiversity Target 11: 17% of terrestrial areas […] are conserved through […] systems of protected areas and other effective area-based conservation measures (Convention on Biological Diversity, 2010). The current situation is particularly alarming because of the continuous expansion of the agricultural activities on these ecosystems (Chebez, 2009; Grau et al., 0000).

At a smaller scale than the one used on this study it is likely to find greater heterogeneity and fragmentation of suitable habitats. The probability of occurrence of the species is likely to be much lower in some sites after considering the effect of local variables describing the species microhabitat (e.g., availability of favorable nesting sites), barriers to dispersal, human impact and biotic interactions. For example, despite high and homogenous probabilities of occurrence predicted for Sierra de las Quijadas National Park (which emblem depicts a Chaco tortoise), the populations in the park are confined to restricted areas (A Ruete & G Leynaud, pers. obs., 2014). This local heterogeneity is likely due to the wide variability in habitat types and geographic accidents as well as due to the pressure of uncontained grassing activity by neighbor’s cattle. Conversely, in Santiago del Estero Province (at the core of the species distribution) the rate of extraction from unprotected areas does not seem to have decline over time (Prado et al., 2012).

This study does not aim to identify healthy populations but to guide the search for them. Also, the predictive maps produced provide a baseline to identify areas where directed strategic conservation management actions (e.g., creation or expansion of protected areas) would be more efficient for this and other associated species. However, before new protected areas or management plans could be delineated, more research is required to better understand what variables drive local habitat selection within areas where the species is expected to be present.

Supplemental Information

Supplemental Information 1 Supporting Information

Appendix S1: Bayesian spatially expanded logistic (BSEL) model and model selection procedure Table S1: Complete list of observations and sources. Table S2: Explanatory variables and model selection. Fig. S1: BSEL model uncertainty Table S3: Presence of Chelonoidis chilensis on protected areas in Argentina and Bolivia.

Click here for additional data file.

We thank two reviewers and the editor for valuable comments on the manuscript. AR thanks the Master Program on Wildlife Management (Universidad Nacional de Córdoba) together with the US Fish and Wildlife Service (USFWS).

Additional Information and Declarations

Competing Interests

Author Contributions

The authors declare there are no competing interests.

Alejandro Ruete conceived and designed the experiments, performed the experiments, analyzed the data, contributed reagents/materials/analysis tools, wrote the paper, prepared figures and/or tables, reviewed drafts of the paper.

Gerardo C. Leynaud conceived and designed the experiments, contributed reagents/materials/analysis tools, wrote the paper, reviewed drafts of the paper.

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
