# Peer review of "Identification of limiting climatic and geographical variables for the distribution of the tortoise Chelonoidis chilensis (Testudinidae): a baseline for conservation actions"

_PeerJ, doi:10.7717/peerj.1298_

## Round 0.1 · original submission · Major Revisions

The findings of your paper have been deemed relevant and methodologically sound, however please pay particular attention to Reviewer 1's comment, that the paper should be far better contextualized for a global readership, and the request from Reviewer 2 that the paper requires a thorough technical and language edit.

(The caption of) Table 1 should say/include that these terms/variables are in addition to the variable water availability.

Reviewer 1 ·

Basic reporting

This is an interesting and well written article describing the current and potential distribution of a turtle species. I think that the article could be a contribution to the literature, but several aspects need to be fixed before it can be published. My main concern is about how the paper is presented. I think that the authors need to do a better job at putting this Article in a broader picture. Clear examples of this are how the paper starts and end. it ends with this sentence "The opposite is also true for t = 0.6.". It would be good if you finish it with some general comments or recommendations. It does not start with the big picture (global decline of herps?) but with a comment on the studied species. there are a few million species out there, and most are vulnerable, should we have papers like this on all these species? I am not saying here that the paper has no value, it has, and a lot. But it needs to be better framed to be useful also for other species and regions of the world.
what is the evidence that this species is in danger? Is there evidence of a population decline?

I have read other paper by the authors and I am positive that they can put this paper in a bigger context.

Experimental design

The study seems to be conducted in a very rigorous way, and the analytical methods are very good.

Validity of the findings

Results are robust. Some comments on aspects marked in the sections "general comments" and in "basic reporting" may make these results even more robust.
It is nice that all the data is available.

Additional comments

Major comments (See also Basic reporting): this species seems to be used as a pet, so there may be a large effect on that on their population. This use may be negative or positive for population growth (e..g for dogs and cats is has not been negative). Also this may affect the current distribution found by the authors (people move it). More information on this is needed.

minor comments:
Please change "fundamental niche" in the abstract. this term is too loaded and is not clear in this context.
What is the evidence that the species cannot pass the Andes? is it too cold?

Reviewer 2 ·

Basic reporting

1. I believe the manuscript should be published as a Short Communication since the scope of the work is relatively contained and has a very focused application.
2. The manuscript needs extensive technical editing and will benefit from a review by someone whose first language is English.
3. The supporting information and maps are sufficient, however, a map depicting the protected area network of the study area would be useful. However, bearing in mind the scale of the study area, protected areas may "disappear" against the backdrop.

Experimental design

1. It is my opinion that the authors have conducted their investigation and have done their modelling with sufficient detail, however, spatial modeling is not within my field of expertise and I would have to rely on another reviewer to comment on the technical aspects and accuracy of the manuscript.

Validity of the findings

1. I found the results and discussion appropriately stated, however, there are a number of statements or descriptions or deductions that need better explanations, or need to be better circumscribed (indicated).

Additional comments

It would be very useful if the authors could spend time and effort on comparing the predicted presence of the tortoise in close proximity of the protected areas (especially those with positive reports) within the eco-regions and indicated spatially how and where these protected areas could potentially be expanded to include critical habitat as pointed out by the model - this is ultimately one of the objectives of the study. Potential expansion of the conservation estate would not necessarily mean purchasing of new land but could also include the voluntary inclusion of private and communal (or state land) land containing healthy tortoise populations into some kind of conservation stewardship agreements with neighbours. This would add value to the predictive value of the model, given that one of the aims is to guide further surveys to discover more healthy outlying populations and to promote their conservation.

Annotated reviews are not available for download in order to protect the identity of reviewers who chose to remain anonymous.

---

## Round 0.2 · accepted · Accept

Thank you for your revised manuscript which has been much improved. I am happy with the decision to move model development details to supplementary material.